# Image Quality Improvement Techniques and Assessment Adequacy in Clinical Optoacoustic Imaging: A Systematic Review

**DOI:** 10.3390/bios12100901

**Published:** 2022-10-20

**Authors:** Ioannis Dimaridis, Patmaa Sridharan, Vasilis Ntziachristos, Angelos Karlas, Leontios Hadjileontiadis

**Affiliations:** 1Department of Electrical and Computer Engineering, Aristotle University of Thessaloniki, 54124 Thessaloniki, Greece; 2Chair of Biological Imaging, Central Institute for Translational Cancer Research (TranslaTUM), School of Medicine, Technical University of Munich, 81675 Munich, Germany; 3Institute of Biological and Medical Imaging, Helmholtz Zentrum München, 85764 Neuherberg, Germany; 4Munich Institute of Robotics and Machine Intelligence (MIRMI), Technical University of Munich, 80992 Munich, Germany; 5German Centre for Cardiovascular Research (DZHK), partner site Munich Heart Alliance, 80636 Munich, Germany; 6Clinic for Vascular and Endovascular Surgery, Klinikum rechts der Isar, 81675 Munich, Germany; 7Department of Biomedical Engineering, Khalifa University, Abu Dhabi P.O. Box 127788, United Arab Emirates; 8Healthcare Engineering Innovation Center (HEIC), Khalifa University, Abu Dhabi P.O. Box 127788, United Arab Emirates; 9Signal Processing and Biomedical Technology Unit, Telecommunications Laboratory, Department of Electrical and Computer Engineering, Aristotle University of Thessaloniki, 54124 Thessaloniki, Greece

**Keywords:** photoacoustics, molecular imaging, clinical imaging, biomedical imaging, image quality

## Abstract

Optoacoustic imaging relies on the detection of optically induced acoustic waves to offer new possibilities in morphological and functional imaging. As the modality matures towards clinical application, research efforts aim to address multifactorial limitations that negatively impact the resulting image quality. In an endeavor to obtain a clear view on the limitations and their effects, as well as the status of this progressive refinement process, we conduct an extensive search for optoacoustic image quality improvement approaches that have been evaluated with humans in vivo, thus focusing on clinically relevant outcomes. We query six databases (PubMed, Scopus, Web of Science, IEEE Xplore, ACM Digital Library, and Google Scholar) for articles published from 1 January 2010 to 31 October 2021, and identify 45 relevant research works through a systematic screening process. We review the identified approaches, describing their primary objectives, targeted limitations, and key technical implementation details. Moreover, considering comprehensive and objective quality assessment as an essential prerequisite for the adoption of such approaches in clinical practice, we subject 36 of the 45 papers to a further in-depth analysis of the reported quality evaluation procedures, and elicit a set of criteria with the intent to capture key evaluation aspects. Through a comparative criteria-wise rating process, we seek research efforts that exhibit excellence in quality assessment of their proposed methods, and discuss features that distinguish them from works with similar objectives. Additionally, informed by the rating results, we highlight areas with improvement potential, and extract recommendations for designing quality assessment pipelines capable of providing rich evidence.

## 1. Introduction

Optoacoustic imaging, also termed photoacoustic imaging, is an emerging, hybrid technology that enables non-invasive visualization of tissue morphological and functional characteristics at depths of up to several centimeters [1]. Non-invasive optoacoustic imaging is based on the optoacoustic effect (Figure 1), in which an instantaneous optical excitation of tissue (usually by means of a pulsed laser) causes the thermoelastic expansion of light-absorbing biomolecules and the subsequent generation of wideband pressure waves that are recorded by ultrasonic transducers positioned on the tissue surface [2]. This hybrid approach leverages and integrates the desirable characteristics offered by pure optical and ultrasonic methods, i.e., optical contrast and acoustic resolution, respectively. Mathematical inversion of the acquired signals renders planar or volumetric images of the initial spatial distribution of acoustic pressure, which is proportional to the absorbed optical energy [3]. Illumination with specific wavelengths enables targeting chromophores of interest, such as oxygenated or deoxygenated hemoglobin, melanin, lipids, and water [4]. In principle, quantitative imaging of chromophore concentrations is possible with multi-wavelength measurements [5], making (‘multispectral’ or ‘spectral’) optoacoustic imaging attractive for an expansive range of clinical [4] and preclinical [6] applications, including, inter alia, histology, dermatology, endocrinology, vascular imaging, and imaging of cancer and inflammation.

Current optoacoustic imaging systems can be broadly categorized as microscopic, mesoscopic, and macroscopic/tomographic, depending on the targeted tissue penetration depth and resolution (Figure 1). Optoacoustic microscopy systems typically employ raster-scanning imaging heads that accommodate a single acoustic detector, and are further classified into optical-resolution (OR-PAM, optical-resolution photoacoustic microscopy) and acoustic-resolution (AR-PAM) categories, depending on whether the achievable resolution is limited by optical or acoustic diffraction, respectively [7]. OR-PAM systems depend on tightly focused illumination to achieve very high-resolution imaging of superficial structures, whereas AR-PAM systems enable relatively deeper penetration at reduced resolution. Mesoscopic systems, such as raster-scan optoacoustic mesoscopy (RSOM), further trade resolution for depth, with wide field illumination and either single or arrays of wideband, focused detectors [8]. Macroscopic implementations also rely on wide field illumination and allow for a significant increase in penetration depth through narrowband detector arrays, at the expense of resolving power [9].

With its strong ability to provide label-free imaging of the endogenous optical contrast of tissue, optoacoustic imaging promises great value for clinical use. Microscopic systems are naturally applicable in histological imaging scenarios [10,11,12], whereas both microscopic and mesoscopic configurations are effective tools for imaging the skin and revealing indicators (inter alia, characteristics of the microvasculature, melanin, or lipid content) of dermatological conditions [4,13,14]. Macroscopic systems, such as the monochromatic optoacoustic tomography (OAT) or the multispectral optoacoustic tomography (MSOT), have great clinical potential for applications including cancer, vascular imaging, imaging inflammation, imaging of lipids/adipose tissues, and imaging of endocrine disorders [15,16,17,18,19,20,21,22]. Examples of optoacoustic images obtained in clinically relevant settings are shown in Figure 2.

**Figure 1 biosensors-12-00901-f001:**
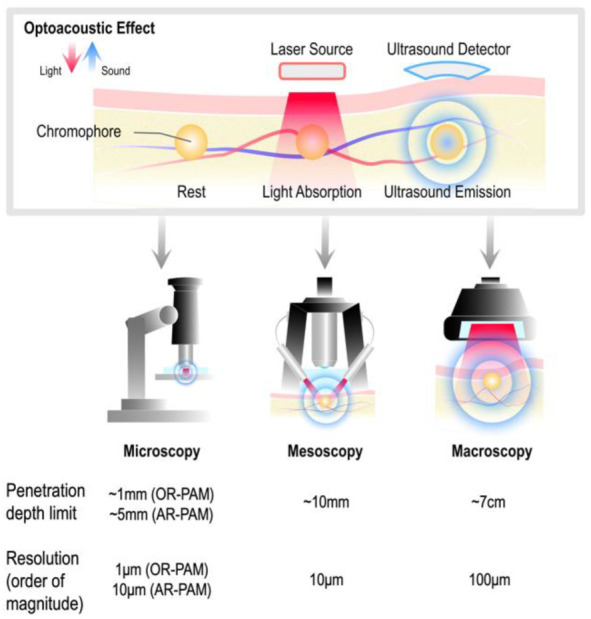
Optoacoustic principle and imaging configurations. **Above**, the optoacoustic effect, in which light absorbed by chromophores in tissue results in emission of ultrasound. **Below**, categorization of optoacoustic imaging configurations into microscopic, mesoscopic, and macroscopic, with the corresponding tissue penetration depth limits and resolution orders of magnitude, as reported in relevant reviews [8,9,23].

Many factors reduce or limit the quality of optoacoustic images, which, in turn, hinder the modality’s potential for clinical translation. These factors may emerge from the imaging hardware [25,26,27,28,29,30], the inexact or approximative image reconstruction algorithms [25,26,31,32,33], the attenuative properties and inhomogeneous nature of tissue (light-tissue interaction phenomena) [34,35,36,37,38,39], or particularities of the acquisition procedure [40,41], and manifest as image noise, artifacts, and poor overall image fidelity.

Naturally, research activity in the fields of denoising and reconstruction quality enhancement has generated diverse approaches in all stages of the imaging pipeline, i.e., prior to, during, and after tomographic reconstruction. For such approaches to become part of what will eventually be standard technical procedures for clinical optoacoustic imaging, quality assessment is necessary. Quality assessment, or quality evaluation, is an important topic in medical imaging that aims to provide means for the performance evaluation of imaging systems and image restoration or compression algorithms [42]. Establishing well-designed image quality assessment methodologies enables impartial comparison of heterogenous image quality improvement techniques in the context of practical expectations, whereas identifying their strengths and limitations helps to establish trust and facilitates progress towards image quality enhancement and, finally, clinical translation.

A common initial step in the process of testing imaging technologies is validation in controlled settings, i.e., via numerical (computational) or physical imaging phantoms that are designed to replicate imaging scenarios of varying complexities; such techniques have been commonly employed in optoacoustic imaging [43]. Numerical phantoms are used in digital simulation environments that offer convenience and flexibility in modifying parameters of the imaging system (e.g., properties of the ultrasonic detector) and the imaged targets (e.g., optical/acoustic properties of absorbers and surrounding media) that may affect image quality [25], whereas physical phantoms provide more realistic testing conditions at the expense of flexibility. Preliminary testing with phantoms enables the assessment of imaging performance in terms of different, isolated characteristics of interest; a fundamental example relevant to optoacoustic imaging is the characterization of image resolution and sensitivity at various depths, by imaging thin absorbers embedded in a typically homogeneous medium, e.g., light absorbing wires inserted in porcine gelatin [44,45]. Additionally, absorbers arranged in a grid pattern can be used to measure geometric accuracy and identify distortions [31], as well as to examine the uniformity of intensity across the field of view. Certain properties of simple phantoms, such as high optical absorbance of targets and low optical and acoustic attenuation of the enclosing medium, oversimplify imaging, and could, therefore, lead to falsely optimistic estimates of imaging performance. To this end, biologically realistic phantoms have been proposed [46,47]. These are fabricated using material that resembles real tissue in terms of morphology and acoustic and optical properties, providing a more complex and challenging testing environment.

Though testing in such controlled environments facilitates quantitative benchmarking and provides valuable insights, it cannot possibly substitute for evaluation with images obtained in real-world clinical scenarios. An image denoising algorithm, for example, could achieve excellent results in images of simply structured phantom targets, but perform poorly in the presence of multiple unpredictable real-world variables, such as subject motion, reflections of out of plane absorbers, and inter-subject variability in anatomy or in other properties of tissue. Thus, during the assessment of image quality improvement techniques, the ultimate benchmark should be their effect on clinical utility, i.e., their capacity to reveal additional, clinically relevant content (e.g., known anatomic landmarks, characteristic biomarkers of disease, or other measurable features related to tissue function, such as blood oxygenation) in scans of human subjects in vivo. Proof-of-concept-level evidence of image quality improvement can be derived with one or few subjects [48,49], whereas more mature and quantitative evaluation can be conducted by involving larger subject groups and performing statistical analyses [50,51].

Although a literature review on signal and image processing methodologies in optoacoustic imaging was recently published [52], the focus was not on studies that involve human subjects, and the matter of quality assessment has not been considered. Within this scope, we conducted a systematic literature search to identify existing approaches to optoacoustic image denoising and quality improvement that have been evaluated on human subjects in vivo. As a first natural outcome, a concise summary of factors that limit optoacoustic image quality, as well as their manifestations in the image domain, is drawn based on the identified body of study material. Secondly, a methodological overview is given, offering a technical outline of every individual approach, and a categorization into signal domain, image domain, and reconstruction or hybrid methodologies. Moreover, to deviate from the simple survey paradigm and provide additional value for upcoming research efforts, we identify subgroups of approaches to improving optoacoustic image quality with a common purpose and similar evaluation procedures, and critically analyze the subgroups’ efficacies; in each subgroup, the individual studies are comparatively rated according to a set of criteria designed to capture important aspects of optoacoustic image quality assessment. The individual criteria ratings are combined to yield comprehensive total ratings, which, after normalization by the subgroup mean rating, resulted in a final ranking of the studies. This enabled identification of the most effective approaches to evaluating optoacoustic image quality to inform future work. In parallel, a broader examination of the distribution of individual criteria ratings revealed limitations and areas of potential improvement in the assessment of optoacoustic image quality. Finally, the primary findings are summarized in a set of recommendations for the comprehensive evaluation of optoacoustic image quality improvement approaches.

To the best of our knowledge, this is the first work to combine a technical overview with a critical, in-depth analysis of reported quality assessment procedures while having evaluation on humans as a prerequisite in optoacoustic imaging.

## 2. Image Quality Limiting Factors in Clinical Optoacoustics

In an ideal imaging scenario, a motionless subject in an acoustically and optically homogenous and lossless medium would be sufficiently illuminated by a high-power light source and fully encompassed by densely arranged ultrasound detectors with sufficient bandwidth and detection angle. In practice, however, these conditions are not met. Image reconstruction algorithms must, therefore, account for and be robust to a variety of limiting factors, while also meeting the demand for real-time operation. Simplifying assumptions allow for shorter image formation times, yet limit reconstruction accuracy, establishing a tradeoff between these two highly important requirements in the clinical setting. To reduce the data acquisition times and hardware costs, sparse acquisition schemes have been considered [53], further limiting the applicability of conventional reconstruction algorithms that depend on complete measurement data. In this section, we identify and describe the limiting factors that contribute to image quality deterioration in optoacoustic imaging. We consider (background) noise to be any image content that does not correspond to optical absorption within the imaged plane or volume. Table 1 gives an overview of the identified limitations and their corresponding manifestations in the image domain.

### 2.1. Limitations in Hardware and Algorithms

In this section, an overview of the limitations in hardware and algorithms is presented and schematically illustrated in Figure 3, where Figure 3a denotes the original (unaffected) image.

High optical irradiance can damage absorbing tissue due to excess heat. Therefore, illumination intensity is limited by safety regulations. The faint excitation of absorbing molecules aggravates the effect of electronic and thermal background noise on the detected signals [30] (Figure 3b).

A homogenous medium, such as coupling gel or water, is commonly used to achieve acoustic coupling between tissue and the imaging probe. The extent of acoustic refraction at the interface will scale depending on the degree of mismatch in acoustic properties thereof, distorting the propagation path of pressure waves, and consequently introducing significant deformation (Figure 3c) in images obtained with conventional acoustic inversion algorithms that assume uniform speed of sound (SoS) [31].

Optoacoustic signals are broadband in nature [54]. The frequency of the emitted pressure waves is inversely correlated to the volume of the absorbing structure. The temporal response of ultrasound detectors used in practice is band-limited; thus, the spectrum of the acquired signals is incomplete: a low-frequency transducer will mostly capture signals corresponding to larger structures, whereas a high-frequency transducer will primarily record components corresponding to finer structures or sharp boundaries [25,28]. Insufficient bandwidth coverage can, therefore, lead to loss of fidelity in the form of missing information in the reconstructed images (Figure 3d).

The transducer arrays used in practice typically resemble an open surface with partial access to the imaged volume, receiving signals only from a restricted field of view, with the exception of circular arrays that may achieve full-view coverage, but can presently only be found in preclinical devices. Individual transducer elements also exhibit direction-dependent sensitivity that can be especially relevant in the near field. These characteristics constitute the limited-view problem, which primarily manifests itself as the deformation of object shapes and incomplete rendering of structures that are poorly visible from the detection surface [25,27] (Figure 3e).

The incomplete acquisition of pressure data in limited-view and limited-bandwidth conditions, inaccurate descriptions of acoustic wave propagation and detection resulting from simplifications employed in reconstruction algorithms, as well as the attenuative and inhomogenous nature of the acoustic medium, may additionally lead to the formation of images with negative values [29,33,55]. Such values are unnatural, as there is no physical interpretation for negative optical absorption. Negative values tend to form in the vicinity of dominant absorbers (Figure 3f), and may completely conceal weaker absorbers nearby.

The Nyquist criterion dictates the appropriate temporal and spatial sampling rates for pressure measurements. Practical constraints due to the prohibitively high cost of densely populated transducer arrays or long acquisition times justify the acquisition of measurements at sub-Nyquist rates, giving rise to aliasing artifacts (Figure 3g). The universal back-projection (UBP) algorithm has also been linked to such artifacts [26].

A common simplifying assumption found in reconstruction algorithms is that transducer elements are point-like. In reality, the detectors have finite dimensions and integrate the pressure over their surface, which is expressed in the point spread function (or spatial impulse response) of the system. When unaccounted for, the finite detector aperture leads to spatially smeared reconstruction of optoacoustic sources [32] (Figure 3h), reducing resolution.

### 2.2. Limitations in Tissue

In this section, an overview of the limitations in tissue is presented and schematically illustrated in Figure 4, where Figure 4a denotes the original (unaffected) image.

The attenuation of light due to optical absorption and multiple scattering events is a major issue in optoacoustic tomography [37], resulting in low and inhomogenous fluence at depth. The waveforms emitted by scarcely illuminated absorbers deep in tissue are fainter (Figure 4b) and, thus, are also more susceptible to background interference, in comparison to those situated closer to the surface. The spatial inhomogeneity of light fluence prevents accurate reconstructions of initial pressure distributions, whereas the non-linear dependency of light fluence on the optical wavelength further impedes quantitative recovery of chromophore concentrations [5].

The attainable resolution at depth is limited by frequency-dependent acoustic attenuation, caused by dissipation and scattering [34], and primarily affecting high-frequency signals. In addition to having lower amplitude, the attenuated waveforms broaden as their bandwidth becomes narrower [35], and the corresponding image features appear blurred (Figure 4c).

The acoustic heterogeneity of tissue also obstructs accurate optoacoustic image reconstruction. The boundaries between tissues of highly contrasting acoustic properties (e.g., bone surfaces) reflect incident waveforms back towards the imaging probe, generating reflection artifacts [36] (Figure 4d), which may overlap with other image features of interest, or be mistaken as regions of actual optical absorption. Unknown variations in acoustic speed also challlenge reconstruction algorithms that naively assume constant SoS inside tissue, distorting the reconstructed images [39].

The inevitable excitation of strong optical absorbers outside the imaged plane or volume (e.g., melanin in the skin) generates intense pressure waves that reach the transducer array either directly or after being reflected by echogenic structures inside the tissue, causing the formation of clutter [38] (Figure 4e) in the reconstructed images. Clutter may significantly obscure signals originating from weakly illuminated absorbers and contribute to misinterpretation.

### 2.3. Limitations in the Acquisition Process

Averaging techniques are regularly used to suppress random noise or artifacts in optoacoustic images. Following multiple excitations and data acquisitions in a short time window, averaging may be performed either in the signal domain or in the image domain [52]. A sequential acquisition scheme is also relevant in raster-scan imaging setups, where multiple images or volumes at adjacent locations are spatially merged [49]. This introduces an additional source of noise, i.e., unintended motion. Fine subject motion due to breathing or arterial pulsation, as well as relative displacement between the probe and the tissue owing to movement of the patient or the operator, will result in misaligned acquisitions and blurred image features [41] (Figure 4f).

### 2.4. The Need Targeted by the Present Review

Clearly, exploring the research efforts which are oriented towards mitigating such limitations is meaningful. Additionally, with clinical utility in scope, there is value in analyzing the individual works and revealing well-evaluated approaches to optoacoustic image quality improvement, which can be considered as more mature in the path towards clinical application. This may also yield valuable insights on proper practices for evaluating optoacoustic image quality.

## 3. Material and Methods

### 3.1. Systematic Literature Search

We conducted a systematic search in PubMed, Scopus, Web of Science, IEEE Xplore, ACM Digital Library, and Google Scholar, for articles published from 1 January 2010 to 31 October 2021. The searched information fields included publication titles, abstracts, and keywords. A comprehensive search query consisting of three main clauses was used: (photoacoustic* OR “photo-acoustic” OR “photo-acoustics” OR optoacoustic* OR “opto-acoustic” OR “opto-acoustics”) AND (reconstruct* OR denois* OR noise OR artifact* OR artefact* OR clutter) AND (clinic* OR “hand-held” or handheld OR “hand held” OR freehand OR “free-hand” OR “free hand” OR human OR humans OR patient OR patients OR volunteer OR volunteers OR subject OR subjects OR individual OR individuals OR participant OR participants OR man OR men OR woman OR women OR person OR people OR “in-vivo” or “in vivo” OR experiment*).

In Google Scholar, publication titles were searched with the first two clauses only, due to its limited functionality. Records of ineligible publication types, such as book chapters or academic theses, were manually excluded. Offline inspection also revealed numerous falsely identified records whose retrieved information did not satisfy all search clauses. Such records were automatically identified and excluded from screening. This automatic process was only applied to records with available abstracts. Records whose abstracts were not retrieved were handled separately. The removal of duplicate records was performed in a semi-automated manner. Groups of duplicate records were automatically identified and subsequently manually inspected to eliminate false positives. The titles and abstracts of the remaining records were screened, and potentially eligible articles were sought for retrieval and full-text evaluation according to the following four inclusion criteria:(i)The proposed method is employed in an imaging scenario.(ii)Image quality improvement or noise reduction is the primary objective.(iii)The proposed method functions entirely after the acquisition of optoacoustic data.(iv)Evaluation of the proposed method with human subjects in vivo is reported.

To eliminate potential bias during the selection of articles, the title and abstract screening process was performed independently by two authors (ID, LH). The same individuals collaborated in the full-text evaluation of the inclusion criteria. All disagreements were resolved by discussion, until consensus was reached. Forty-five eligible papers were finally selected, including one that was not retrieved by the search, but was identified in the references of a related, included work. An overview of the search and screening process is given in Figure 5, which depicts the corresponding PRISMA 2020 diagram [56].

### 3.2. Subgroup Analysis, Rating Criteria, and Procedure

For the systematic assessment of the included studies in terms of evaluation adequacy, a comparative rating approach was adopted. Initially, four authors (ID, AK, PS, LH) collaborated in the conceptualization and design of the rating process. A focus group of three authors (ID, AK, LH) executed the rating process. In the following, all papers were cooperatively evaluated by all members of the focus group. All disagreements were resolved by discussion, until consensus was reached.

First, studies were classified according to their general motivation, forming subgroups of studies with a common purpose (i.e., intention to solve a common problem) and evaluation procedures with comparable characteristics. The subgroup analysis was essential to enable fair comparison, since the studies exhibited significant variety in the design of evaluation experiments. Thirty-six out of forty-five studies were finally included in the subgroup analysis; the remaining nine studies were not included either due to their purpose not suiting any subgroup, or due to the impossibility of meaningful comparison to the other studies of the subgroup. The subgroups are summarized in Table 2.

Following the assignment of studies to the subgroups, the evaluation section of each paper was thoroughly analyzed, considering, in detail, the design and results of all reported experiments on numerical and physical phantoms, as well as on humans or animals in vivo. Through this analysis, a set of criteria was established, according to which the studies were compared and rated. The criteria, described in Table 3, aim to comprehensively capture the important aspects of the evaluation procedure in an organized and analytic manner.

The rating procedure was independently conducted for each individual subgroup. Initially, a subset of applicable criteria was selected according to the specific characteristics of the included papers. In particular, C_2_ is only applicable in the S_RES_, S_SPR_, and S_LVB_ subgroups, where reference methods could be identified in preceding research, and C_6_ is only applicable in the S_SPR_ subgroup. The remaining criteria are applicable to all subgroups.

In all subgroups, except for S_MOT_, the experiments reported in each study were classified into three categories, involving numerical, phantom, and human subjects. Experiments involving animals were reported in only two studies [48,61], and added little new information to the evaluation; these experiments were, therefore, considered as complementary to studies with human subjects, and assigned to the category of experiments with human subjects. Each category of experiments was analyzed according to the applicable criteria, and ratings were given to each criterion according to the following scale: 0: absent, 1: lacking, 2: adequate, 3: ample, 4: thorough. The minimum possible increment was 0.5. A fundamental motivation behind the rating assignment procedure was to reflect the relative quality, with respect to each criterion, of the studies included in each subgroup. Therefore, it is more appropriate to consider the ratings as being relative, not absolute. The ratings given for each criterion were combined into a subtotal rating for each category of experiments, via weighted addition. All criteria contributed equally to the subtotal with a weight of 1, except for C_6_, which was empirically assigned a weight of 1/5 to reflect a smaller importance; it was generally considered as being a secondary criterion. The subtotals were combined into a total rating via weighted addition, followed by division with the sum of the weights of the applicable criteria, as shown in Figure 6. Subtotals corresponding to numerical and phantom experiments were empirically assigned a weight of 1/3 to emphasize the significance of experiments on human subjects.

The S_MOT_ subgroup was handled slightly differently, as the studies primarily performed their evaluations via experiments with human subjects, i.e., the most appropriate way to draw useful conclusions in such a setting. In some cases, synthetic motion was added to scans obtained from either physical phantoms or human subjects for preliminary validation of the proposed methods. Such experiments were taken into consideration, but were not analyzed into separately rated criteria (as in the other subgroups) due to the simplicity they exhibited. Instead, they were handled as a single criterion, i.e., they were given a single rating and contributed to the total rating equally to the other criteria.

The ultimate objective of the rating procedure was to highlight studies that conducted, in their subgroup, relatively comprehensive and unbiased evaluations of their proposed methods for the improvement of optoacoustic image quality. To achieve this, the total rating was divided by the mean rating of the corresponding subgroup, yielding the normalized total rating, a metric of the deviation from the subgroup average. This metric was finally used to select a set of studies that surpass the threshold defined by the upper quartile (Q3) of the normalized total ratings distribution. An illustrative overview of the rating procedure is given in Figure 6.

## 4. Results

### 4.1. Signal Domain (Pre-Processing) Approaches

Table 4 summarizes the methodological overview of the identified signal domain approaches. The detailed description follows.

In an effort to correct for incomplete structure reconstruction and suppression of weak absorbers in the vicinity of dominant boundaries, caused by the limited-view and -bandwidth conditions, Kruizinga et al. [27] developed an ultrasound-guided technique. Based on the observation that optoacoustic and ultrasound images share common morphological information, a co-registered ultrasound image is manually segmented into relevant structures, which are assigned initial source pressure values derived from reconstructed optoacoustic images, and used to simulate the full optoacoustic wave field. The simulated field is used to mitigate the limited-view errors by completing missing wavenumbers in the recorded signals, or to reveal fine structures by masking out dominant boundary signals.

Wang et al. [82] proposed a multi-sample averaging approach to improve the inherently low signal-to-noise ratio (SNR) of acquired optoacoustic signals. The cross correlation between acquired frames is calculated to determine relative time shifts and adaptively align the waveforms. This step is preceded by upsampling via cubic spline interpolation for greater precision. The aligned signals are finally averaged, suppressing random noise and enhancing detail. The aforementioned procedure is also the ultimate step of the method reported by Cao et al. [83], in which the frequency response of the transducer is used to design an appropriate transfer function. The latter is utilized in an inverse signal filtering process with Wiener deconvolution to amplify frequency components that were suppressed due to limited transducer bandwidth.

Hill et al. [72] utilized Singular Value Decomposition (SVD) to obtain a sparse representation of the external noise induced from the laser electronics in the excitation source. Such noise manifests itself as planar patterns that are consistent across the data vectors acquired in parallel from multiple transducer elements. The noise was assumed to be additive, modelled with an experimentally determined number of Singular Value Components (SVC) and subtracted from the data matrices prior to image reconstruction. Averaging of multiple denoised images, followed by envelope detection with the Hilbert transform, produced the final images.

In the work by Nagaoka et al. [68], it is assumed that the original optoacoustic waveform generated from a small point or cylindrical source approximately resembles a unipolar pulse. However, propagation in acoustically lossy tissue and the limited detector bandwidth distort the recorded waveform, introducing spurious time side lobes that reduce axial resolution. To alleviate the distortion, the signals received by each element of the transducer array are subject to Wiener filtering for bandwidth restoration, followed by phase correction filtering and thresholding. The corrected signals are used to reconstruct the image with a delay-and-sum method. The negative amplitudes in the reconstructed image are eliminated by thresholding.

A motion correction algorithm for RSOM was proposed by Aguirre et al. [40], in which a cumulative cross-correlation surface (CS) function is calculated from the acquired 3D sinogram. Broadly, this 2D function measures the cross-correlation between recordings at adjacent scan locations that compose the sinogram. Its continuity is assumed to reflect the disruptions in the sinogram caused by motion. The calculated CS function is compared to a synthetic smooth CS function to quantify the vertical displacements between the detector and the skin at each scan location caused by breathing, heart beating, and arterial pulsation. The displacements are then accounted for during image reconstruction.

Schwarz et al. [49] also developed a motion correction algorithm for RSOM. Vertical movement between the melanin layer at the skin surface and the detector is expected to cause matching disruptions in the acquired 3D sinogram. The algorithm works by first detecting the skin surface at each scan point of the raw sinogram. The detection approach differs based on whether the imaged region lies in an area of hairy or hairless skin. The estimated discontinuous 3D surface is subsequently corrected with a moving average filter, resulting in a smooth surface. Finally, for image reconstruction, the position of the detector is adjusted using the relative offset between the two surfaces.

Hu et al. [26] considered the problem of spatial aliasing in full-ring geometry optoacoustic tomography. Separate spatiotemporal analyses are performed to identify concentric, disk-shaped spatial regions, where aliasing may occur either due to insufficient spatial sampling density or due to the characteristics of the UBP reconstruction algorithm. The two conditions are termed as aliasing due to spatial sampling (SS) and image reconstruction (IR), respectively. It is shown that aliasing entirely due to IR can be eliminated by spatial interpolation, i.e., numerically increasing the number of detection elements. Additionally, a radius-dependent temporal lowpass filter is proposed to remove aliasing due to SS. A complementary analysis for the linear array geometry is also reported.

### 4.2. Image Domain (Post-Processing) Approaches

Table 5 summarizes the methodological overview of the identified image domain approaches. The detailed description follows.

Negative values have no physical meaning, and, when originating from large image features, can mask nearby fine structures, rendering them invisible to spectral unmixing techniques. To uncover such structures, the method proposed by Taruttis et al. [33] employs the stationary wavelet transform to decompose the images taken at different wavelengths into separate scales, which are then individually inverse-transformed and spectrally unmixed by a non-negative least squares algorithm. By unmixing each scale independently, interference from structures in other scales is prevented. A separate multiscale image is then formed for each chromophore by adding the different scales together.

To enhance the visibility of vessels in the presence of background noise introduced by limitations of the acquisition system and optical absorption by irrelevant chromophores in an OR-PAM setting, Haq et al. [84] utilized Gabor wavelet filtering, in consideration of the multiscale nature of vessels. Furthermore, a Hessian-based filtering technique classifies structures into tubular, blob, and plate-like, based on the relationship between eigenvalues of the Hessian matrix calculated at each voxel. The Euclidean norm of the matrix is used to differentiate between structures of interest and noise. The latter filtering technique is repeated at different scales, ultimately preserving the maximum response of each scale.

To accurately align multiple shot volumes for the optoacoustic imaging of large body parts in the presence of body motion, Bise et al. [74] proposed to organize the volume images as rows of an observation matrix, and to transform the latter to low-rank by a coarse-to-fine optimization framework. In the coarse step, after noise suppression with a vessel enhancing filter, a multi-scale pyramid scheme is used to find the optimal transform function. In the fine step, the transform function is further optimized, while also taking noise into consideration. Specifically, the coarsely aligned, non-filtered volumes are decomposed into a low-rank vessels foreground, a dense noise background, and a sparse complement component. A statistical prior constraint is introduced in the formulation of the refining optimization process, in which the average of the dense noise background at each spatial voxel is forced to be constant, based on the examination of experimental data.

A deep convolutional neural network architecture consisting of three sequentially connected image enhancement units was developed by Anas et al. [85]. The units are jointly trained on triads of cross-sectional target images from blood-vessel-mimicking phantoms with sequentially increasing image quality: the deeper the unit, the better the quality of the target image. This pattern is assumed to guide feature extraction more effectively across the network. The method was proposed to reduce the need for extensive frame averaging, which would lead to motion artifacts in an LED-based optoacoustic system, where the output power is inherently limited.

A method for the identification and removal of in-plane reflection artifacts was proposed by Nguyen et al. [36], based upon the assumption that a reflection artifact appears deeper than, and its spectral response is principally correlated with, the image feature it originated from. Given the images taken at four different wavelengths, features of the clearest image are detected via segmentation, and their spectral response is estimated using the rest of the images. Then, features with high cross-correlation, in terms of spectral response, are grouped. The dominant feature in each group is assigned as a real absorber, whereas fainter features appearing at larger depths are assigned as reflection artifacts, and are removed by setting the corresponding pixel values to zero. An alternative that does not require segmentation is also given, in which the same correlation analysis is instead performed for each pixel of the image. In a subsequent work [73], the technique was slightly adjusted and shown feasible in a setting where two wavelengths were available instead of four.

Ma et al. [86] developed a processing method for images generated with an OR-PAM variant that employs a micro-electromechanical system (MEMS) scanning mirror to accelerate the raster-scanning procedure. Such a system is prone to distortion and resolution limitations, stemming from electrical, ultrasonic, optical, and thermal particularities of the hardware. The proposed method, termed as spatial weight matrix (SWM), processes the image in three distinct stages: first, an adaptive wiener filter is used for noise reduction. Then, a registration module corrects for geometric distortions introduced by the scanning procedure. Finally, a deconvolution operation accounts for error that originates from neglecting the surface area of the ultrasonic sensor.

Shen et al. [29] studied the formation mechanisms of negativity artifacts, and evaluated the performance of two simple post-processing approaches. In the forced-zeroing approach, negative values are simply set to zero, whereas in the envelope detection approach, the negative components of the image are reverted, and amplitude profiles are extracted via Hilbert transform. The envelope detection approach results in images with broadened features that may, however, retain more information about the geometry of the optoacoustic sources.

In optoacoustic dermoscopy, involuntary subject motion causes vertical displacement, whereas the scanning process itself causes a horizontal misalignment. Vertical displacement can be decomposed into rigid motion, mainly due to sudden movement of the human subjects, and non-rigid motion, caused mainly by breathing, heart beating, and arterial pulsation. Horizontal displacement is considered as rigid motion. The algorithm presented by Cheng et al. [75], based on subpixel motion estimation, corrects for motion in two steps: in the Subpixel Global Motion Correction step, horizontal and vertical rigid displacements between adjacent B-scans are corrected for via a spline interpolation method. In the Subpixel Vertical Motion Correction step, vertical non-rigid displacements between adjacent A-lines of the same B-scan are estimated via a cumulative differential function inspired by the approach by Aguirre et al. [40], and offset.

Erlöv et al. [41] examined the feasibility of using a regional motion correction algorithm for hand-held 2D optoacoustic tomography. In contrast to global motion correction, a regional approach could also account for internal tissue movement. The technique is based on intensity phase tracking (IPT) performed on the interleaved ultrasound images that are co-registered with the optoacoustic images. The use of ultrasound images makes the solution appropriate for multispectral scenarios, where consecutive optoacoustic frames depict different absorber distributions, and, therefore, cannot be used reliably for tracking.

### 4.3. Reconstruction and Hybrid Approaches

In contrast to signal and image-domain approaches, reconstruction approaches address the targeted optoacoustic image quality limitations in an end-to-end manner, integrating the quality improvement techniques in the image reconstruction pipeline. Approaches that incorporate multiple elements, such as analysis in the signal domain, reconstruction, and processing in the image domain, are considered hybrid. Table 6 summarizes the methodological overview of the identified reconstruction and hybrid approaches. A detailed description follows.

#### 4.3.1. Beamforming Approaches

In conventional delay-and-sum (DAS) beamforming, the images are formed by summation, at each spatial location, of the back-projected recorded signals. Back-projection is based on signal time-of-flight calculations and delaying. The signals are expected to add up coherently at the locations they originated from. A quantitative measure of phase alignment between signals is the Coherence Factor [88], calculated as the ratio of the squared coherent sum to the incoherent sum of the signals. The coherent sum is a regular sum, whereas the incoherent sum is defined as the sum of the squared signals.

Wang et al. [70] integrated focal-line-based 3D reconstruction with coherent weighting to improve the elevation resolution of a linear array. In focal line 3D reconstruction, the time of flight of the signals to be added by the DAS algorithm is corrected by considering a propagation path that crosses the focal line of the cylindrically focused transducer element, rather than a direct straight path. Weighting the reconstructed image with the coherence factor of the signals added at each location further enhances the elevation resolution.

Jeon et al. [66] enhanced the delay-multiply-and-sum (DMAS) beamformer with a modified coherence factor to improve resolution and reduce noise by suppressing sidelobes. DMAS beamforming involves the combinatorial coupling and multiplication of the time-delayed recorded signals. Products between in-phase signals have higher magnitude; thus, they contribute more to the final output, i.e., the sum of the products. The coherence factor is employed to weigh the DMAS outputs, in accordance with the coherence of the summed terms.

Mozaffarzadeh et al. [67] acknowledged the inherent existence of DAS procedures in the algebraic expansion of the DMAS beamformer. Considering that the nonadaptive nature of DAS will, therefore, implicitly limit the performance of DMAS, they proposed to modify the latter by replacing the DAS algebra terms with the superior minimum variance (MV) beamformer to reduce sidelobes and improve resolution.

Fournelle and Bost [65] presented a technique to improve the reconstruction of targets resembling point sources. The technique is based on the analysis of the time-delayed signal amplitudes that are summed to reconstruct individual pixels in DAS beamforming. A confidence measure, corresponding to the likelihood of an individual pixel for being an optoacoustic source, is derived by comparing the distribution of measured amplitudes to a theoretical model. The resulting confidence map is multiplied pixelwise with the DAS beamformed image, lowering the amplitudes where the confidence is low. Three candidate confidence parameters with varying degrees of modeling accuracy are experimentally evaluated.

Ma et al. [44] designed a beamforming algorithm to suppress sidelobes and artifacts, and to improve the separability of adjacent structures. In contrast to common beamforming techniques that only calculate a single value for the initial optoacoustic pressure, the proposed method calculates an N-shaped waveform for each pixel of the reconstructed image. The estimated bipolar N-shaped signals are enveloped with the Hilbert transform to obtain positive pulse signals. The location, in the time axis, of the maximum pulse amplitude determines the distance from the considered pixel to the actual location of the acoustic source, and is used to narrow the signal envelope accordingly: a large distance will manifest itself as sidelobes and artifacts; therefore, suppression is required. The initial value of the resulting pulse is finally chosen as the reconstructed pixel intensity.

Yang et al. [51] introduced a lag-based DMAS beamformer with a coherence factor (DMAS-LAG-CF) to enhance resolution and contrast. Lag-based DMAS [89] is a two-stage algorithm that first employs channel-wise correlation of the original input signals to synthesize new signals, which are then input to a filtered DMAS (F-DMAS) beamformer. In the latter, the output of DMAS is band-pass filtered to suppress the DC and higher frequency components [90]. The coherence factor is embodied in the first stage of DMAS-LAG-CF, where it weighs the input signals prior to the correlation operations.

Tordera Mora et al. [69] considered two coherence-based beamforming approaches, i.e., short-lag spatial coherence (SLSC) [91] and F-DMAS, and studied their merits and limitations: the first is robust against noise, but does not preserve relative signal magnitude (which prevents quantitative interpretation), whereas the second preserves signal magnitude, but suffers from reduced contrast. They proposed a mathematical unification of both beamformers into a single equation. The developed generalized spatial coherence (GSC) algorithm aims to address both limitations by combining the technical elements that give each technique its advantages.

#### 4.3.2. Machine Learning Approaches

Hauptmann et al. [59] incorporated learned regularization in an iterative model-based reconstruction approach to improve results in limited-view and sub-sampling scenarios. Each iteration is performed by a separate Convolutional Neural Network (CNN) model that receives two inputs: the current iterate image and the gradient of the data fit term. The latter measures the deviation of the forward model’s output to the observed data, and is typically used to perform updates in a gradient descent scheme. It is argued that the gradient information introduces robustness to perturbations. The CNN models output the next iterate image, and are individually trained to minimize the difference between their output and the ground truth. Segmented vessel images are used to generate synthetic training data, therefore making the approach suitable for imaging vessel-rich targets. Additional fine-tuning with real measurements from human subjects is also considered.

In an identical learned iterative reconstruction framework [58], an approximate k-space (Fast Fourier Transform (FFT)-based) forward model is used to achieve fast reconstruction from sub-sampled data, and to have the CNN learn to correct for the highly structured artifacts introduced to the gradient information by the approximate model. The CNN are initially trained on synthetic data generated by segmented vessel images, and fine-tuned with sub-sampled data from human measurements.

With microvessels as the target of interest, and the limited-view and -bandwidth problems in scope, Kim et al. [79] proposed a reconstruction framework based on the UNET [92] encoder–decoder CNN architecture. The 2D raw data matrix is transformed into a 3D form with two spatial dimensions and a channel dimension, basically consisting of one 2D image slice for each transducer element (channel), or, viewed otherwise, one 1D vector of delayed signals (the propagation delay profile) for each image pixel. This preprocessing is proposed to retain the information in the raw data while providing a convenient input structure to the CNN. The CNN is trained on data synthetically generated from the target vascular images.

To combine the merits of signal and image domain processing, Lan et al. [80] designed a CNN architecture with two inputs: the measured raw data and the corresponding DAS reconstructed image. The features extracted from the inputs via two separate encoder networks are subsequently concatenated and forwarded to the decoder, which produces the output image. Skip connections between layers of the decoder and the two encoders provide an information sharing channel between the component networks. The data and images used to train the network are generated via simulation of limited-view and bandwidth acquisition, based on segmented blood vessel images, which are also used as ground truth targets.

Meng et al. [62] provided an efficient, non-iterative method to improve reconstruction quality in 3D optoacoustic tomography under sparse sampling conditions. The method assumes a scanned acquisition procedure, where successive B-scan frames comprise the 3D volume. One every two or three frames is obtained with fully sampled measurements, whereas the rest of the frames are obtained with sparse measurements. Backprojection is used for the initial reconstructions. The fully sampled frames are then used as training samples to construct a PCA basis, on which the sparsely sampled frames are projected and subsequently recovered from. The PCA basis represents a low-dimensional feature space that describes the principal characteristics of the training data, exploiting useful information across frames.

In a similar direction and experimental setting, Liu et al. [61] significantly reduced the number of required fully sampled frames by incorporating a Dictionary Learning approach in an iterative model-based optimization scheme. Approximately 5% of the B-scan frames are uniformly selected from the imaged volume, and are reconstructed from fully sampled measurements. A dictionary is trained on the fully sampled frames, and is afterwards employed in the objective function with two terms that promote consistency between the image and the dictionary domains and sparsity to reconstruct the remaining undersampled frames. The dictionary provides an adaptive sparse representation mechanism that encodes prior information from the training data.

#### 4.3.3. Compressed Sensing (Sparsity-Based) Approaches

Compressed Sensing (CS) techniques exploit the inherent sparsity that natural signals and images exhibit in certain transform domains, to obtain accurate reconstructions from sub-Nyquist sampled data. A representation vector is considered sparse if it primarily consists of zero elements. Thus, CS can be applied to image reconstruction problems with a constrained optimization approach by searching for solutions that are both consistent with the measured data and sparse in an appropriate transform domain, therefore utilizing sparsity for regularization. Jing et al. [60] applied Total Variation-based CS in both time domain (TD) and frequency domain (FD) reconstruction, and demonstrated the superiority of the FD-CS variant over the TD-CS and conventional backprojection approaches.

Meng et al. [48] reported on a CS reconstruction framework that, besides sparsity, incorporates partially known support (i.e., known nonzero locations in the transformed sparse domain) as an additional prior. The optimal sparse image vector in the wavelet domain is solved via minimization of an objective function whose sparsity-promoting term only penalizes the number of non-zero elements that do not belong to the partially known support. The latter is determined in each iteration by thresholding the sparse vector. This further confines the space of candidate sparse solutions, and results in less undersampling artifacts. An iteratively reweighted conjugate gradient descent technique was developed for minimization.

Han et al. [57] developed a sparsity-based approach for 3D optoacoustic reconstruction, highlighting the suitability of such approaches in the 3D case, where images are naturally more compressible. A sparse representation of the reconstructed image in the wavelet domain is iteratively optimized. The step size for the gradient descent minimization procedure is calculated via the Barzilai–Borwein line search approach, which takes the previous and current solutions, as well as the gradient of the objective function, into account.

Meng et al. [63] implemented an advanced CS reconstruction technique that exploits structural dependencies between wavelet coefficients located adjacently across different subbands and scales, via a Gaussian Scale Mixture model (GSM). The structured sparsity is incorporated as prior information in the objective function by weighting the wavelet coefficients in the sparsity-promoting term. The diagonal weighting matrix is refined on each iteration by filtering the current iterate wavelet coefficient images with a Wiener linear estimation-based GSM (wGSM) operator. The filtering clears the wavelet coefficient images from sparse-sampling artifacts.

Pan et al. [64] provided theoretical evidence supporting the appropriateness of the Curvelet frame as an optimal sparse representation of both the initial pressure distribution image and the volume of recorded pressure data. Building on this, they designed and compared two CS reconstruction approaches, both in a variational framework, representing the solution in the Curvelet basis. Because the latter is overdetermined, they employed an iteratively reweighted l1 minimization process, in which the solution’s coefficients are weighted according to their magnitude, to enhance sparsity. In the first two-step approach, the full volume of optoacoustic data is initially recovered from the subsampled measurements, and the final reconstructed image is then obtained via time reversal. Thus, computationally expensive iterations involving the forward and adjoint acoustic operators are avoided. In the second approach, the image is reconstructed directly from the subsampled data. A non-negativity constraint is considered for the objective function of the latter approach.

#### 4.3.4. Other Approaches to Optoacoustic Image Quality Improvement

To distinguish between signal and clutter, Alles et al. [71] introduced a hybrid method that exploits the relatively lower coherence of clutter signals across channels of the array transducer. SLSC is calculated to quantify the signal coherence in each spatial location, and is used as a weighting map to selectively suppress clutter in reconstructed optoacoustic images. The amplitude-insensitive SLSC map is normalized prior to multiplication with the image in order to minimize distortion of the true amplitude values.

To incorporate the acoustic properties of the imaged object in the reconstruction process, Lutzweiler et al. [77] proposed a segmentation algorithm in the signal domain. The algorithm analyzes characteristic features of the hilbert-transformed unipolar sinogram, to produce, via an iterative optimization process, an SoS map that consists of up to three homogenous SoS compartments separated by convex and smooth boundaries. The derived SoS map is then deployed during image reconstruction. The applicability of the method is limited due to requirements on the number, geometric arrangement, as well as optical and acoustic properties of the compartments. Large angular coverage of the detection surface around the imaged object is also desirable.

Yang et al. [78] studied the impact of assuming uniform SoS during reconstruction in the context of multispectral optoacoustic imaging, and developed a model-based reconstruction technique to account for the acoustic mismatch between the heavy water coupling medium and the tissue, and to eliminate spectral smearing artifacts. Assuming linearity of the wave equation with respect to SoS variation, the pressure distribution is taken as the superposition of two contributing terms, corresponding to the heavy water and the tissue. This is incorporated into the model matrix of the discretized forward expression, which is inverted via regularized optimization.

Also concerned with minimizing image distortion due to SoS variation, Deán-Ben et al. [76] corrected the time of flight of back-projected signals in a volumetric imaging setup: the wave propagation path is estimated, taking acoustic refraction at the interface between the coupling medium and the tissue into account. The propagation is modelled by means of the Fermat’s principle, and an iterative method is used to determine the location of the point of incidence on the acoustic boundary. The application of pre-processing techniques, including impulse response deconvolution and band pass filtering, was also reported.

The simplifying assumption of point-like transducers during acoustic inversion results in the deterioration of reconstruction accuracy. To account for the 3D shape and frequency-dependent directional sensitivity of transducers used in practice, Ding et al. [32] proposed to realistically model the finite size of transducer elements by subdividing the transducer surface into discrete elements. The signal measured by each transducer is, therefore, approximated by the integral of the pressure on the transducer surface. Equivalently, in the discrete forward model, the model matrix is expressed as a weighted (by surface area) sum of multiple model matrices, one for each discrete surface element.

Chowdhury et al. [31] proposed to use the total impulse response (TIR) to model the effects of different system components on image reconstruction. TIR can be decomposed into the spatial impulse response (SIR) and the electrical impulse response (EIR) components, which capture the geometrical and electrical particularities of the system, respectively. They derived the theoretical SIR by mathematically modelling the effect of acoustic refraction at the interface between tissue and the coupling medium. Subsequently, they experimentally obtained the TIR at a few measured locations, combined it with the simulated theoretical SIR to extract the approximate EIR, and finally generated a synthetic TIR (sTIR) by combining the simulated SIR with the derived EIR. The sTIR is incorporated in the forward model in a model-based reconstruction scheme.

Considering that ultrasound images are naturally less prone to the limited-view problem, Yang et al. [81] developed an ultrasound-driven optoacoustic inversion scheme to compensate for the effects of incomplete data acquisition in a limited-view acquisition scenario. Co-registered ultrasound images are segmented into regions of homogenous acoustic properties, which may be reasonably expected to also exhibit homogeneity in terms of optical properties. The segmented images are used to derive a regional Laplacian regularization matrix that incorporates this prior structural information in the model-based inversion process, promoting the reconstruction of images that are smooth inside each such region.

Steinberg et al. [50] demonstrated a real-time reconstruction technique appropriate for limited-view and low-SNR acquisitions. A detailed forward model matrix accounts for the acoustic absorption properties of the medium, as well as the effect of transducer element size, shape, sensitivity, directivity, and impulse response. For physically accurate reconstructions and robustness to noise, non-negativity and anisotropic total variation regularization are incorporated. The notion of mathematical superiorization is used to enhance the performance of a non-linear conjugate gradient algorithm and achieve fast convergence.

Considering that multiple surface measurements could be valuable in addressing a range of limitations, including interference from out-of-plane absorbers and decreased contrast at depth, Zalev and Kolios [87] implemented a framework for simultaneous 3D image reconstruction and probe motion tracking. The process is formulated as a convex optimization problem, in which the reconstructed image, as well as the probe configuration, are jointly solved via a combined minimization objective. The developed system matrix, which models acoustic signal generation and acquisition in the objective function, accounts for wavelength-specific optical absorption, probe position and orientation, radiant fluence distribution, and spatio-temporal transducer impulse response.

### 4.4. Rating and Selection of Studies

Figure 7 synoptically illustrates the distribution of the normalized total ratings against the corresponding box plot, which defines the threshold ($Q_{3}=1.1$) used to select the studies that are finally featured. Points that correspond to ratings above the threshold are colored red. The selected studies are referenced in Table 7, with their subgroups and normalized total ratings. Figure 8 gives an all-encompassing overview of the individual criteria ratings for the human-involving experiments in all studies of all subgroups. The analytical rating tables for all subgroups, experiment categories, and criteria are given in the Appendix A.

## 5. Discussion

### 5.1. Close Inspection of Featured Works

We initiate the discussion with a close inspection of the works that surpassed the selection threshold (Figure 7, Table 7), selecting and mentioning key characteristics that contributed to their deviation from the rest of the studies in the respective subgroup.

The work by Chowdhury et al. [31] stands out in S_ACM_, primarily due to the experiments with both numerical and physical (printed) phantom targets that uniformly cover the entire field of view with point absorbers. Imaging such targets enabled objective baseline validation of the method’s correction capacity, and could constitute a universally applicable preliminary step to evaluate a wide range of quality enhancement approaches. The availability of a reference image for the printed phantom target allowed quantitative assessment of structural fidelity with the Structural Similarity Index Measure (SSIM) [93], applied usually in simulated settings only. Extensive simulations rendered the beamforming approach by Ma et al. [44] distinct in S_RES_; in addition to the typically used arrangement of point targets on the axial direction, a large circular source was simulated to assess shape distortion. Additionally, closely situated numerical and physical point targets were imaged to investigate improvement in the separability of adjacent structures, a commonly overlooked, but important, feature. In S_SPR_, the learning-based reconstruction technique by Hauptmann et al. [59] is also set apart, owing to numerical experiments in which a dataset of vessel-rich volumetric images was generated using a collection of lung CT scans to simulate optoacoustic measurement data. Such data may more closely resemble measurements obtained in real practice, and could be useful for the initial evaluation of various approaches, besides learning-based ones. Another advantage comes from the availability of multiple different images to evaluate with, making statistical quantitative analysis possible.

Two approaches [49,75] were featured in S_MOT_. The OR-PAM motion correction technique by Cheng et al. [75] was validated with a well-balanced set of experiments in both artificial and realistic settings. The former examined the robustness of displacement estimation to synthetically added noise at varying levels, as well as the tolerance to different magnitudes of misalignment. In the latter, both the back of a human hand and a palm were scanned, adding to the variety. The calculation of the PSNR and SSIM metrics between all consecutive pairs of adjacent B-scans provided a measure of inter-B-scan similarity, a useful means to quantify the increase in alignment and its consistency. Three distinct depth layers in the reconstructed volume were visualized separately to demonstrate the correction effects on the vascular networks of three different scales. Resolution was quantified with the full width at half-maximum (FWHM) metric at six randomly selected vessel profiles in each depth layer. This depth-wise qualitative and quantitative assessment showcases diligence and attention to detail, also seen in the evaluation of the RSOM motion correction algorithm by Schwarz et al. [49]. An interesting addition therein is the separation of low- and high-frequency sub-bands in the visualization and calculation of the depth-wise contrast-to-noise ratio (CNR), again facilitating assessment at different scales and depths. Nevertheless, what distinguishes this study further is the examination regarding quantification capacity and clinical utility; first, images of healthy skin and a psoriasis plaque were compared to observe that the broadening of capillary loops, a typical biomarker of psoriatic skin, was only visible after motion correction. In addition, using a multispectral RSOM configuration, blood oxygenation measurements across a single vessel were shown to exhibit less abrupt, more biologically plausible variations that were spatially associated with regions of vessel bifurcation. Though analysis on a single vessel might not be considered sufficient to provide significant evidence, this is a step in the right direction, exploiting the full potential of multispectral optoacoustic imaging.

Among the featured works, those that exhibited exemplar attention to clinical and quantification aspects are discussed in the following. The evaluation of the beamformer by Yang et al. [51] attained the highest rank in S_RES_, with the quantitative analysis in terms of discrimination capacity being a key determining factor; in an experiment involving a cohort of 28 patients, descriptive features of histograms of optoacoustic image regions corresponding to cancerous and non-cancerous ovaries were analyzed to test for a statistically significant difference between the two groups. In addition, a logistic regression analysis on sets of features was performed, and area under the receiver operating characteristic curve (AUROC) metrics were compared between methods. Overall, the proposed method performed favorably. In S_LVB_, attention to clinically relevant aspects also distinguished the two featured studies [50,81]. The prior-integrated reconstruction approach by Yang et al. [81] was first assessed visually by confirming an enhancement in clarity of the radial artery at three different depths, as well as in visibility of the carotid artery lumen and, if present, atherosclerotic plaque. Quantitatively, between groups of three healthy volunteers and five atherosclerotic patients, a statistically significant difference in the lipid content of the carotid region could only be detected with the proposed approach. Lastly, the quantification capacity, in the work of Steinberg et al. [50], was initially evaluated in vitro with multispectral measurements of an indocyanine green (ICG) tube in chicken breast tissue, demonstrating that sufficient correlation between the measured and the known reference ICG spectra could only be achieved using the proposed reconstruction technique. In the clinically relevant in vivo assessment, ten patients with suspected prostate cancer lesions were scanned with multiple optical wavelengths before and after injection of ICG. A statistically significant difference in optoacoustic amplitude in the prostate region, as well as a dependence between relative optoacoustic amplitude and ICG dose, was only observable in images reconstructed with the proposed method.

### 5.2. Broader Assessment of the Analyzed Works Per Individual Criteria

Following the inspection of notable details in the featured studies, a broader assessment through the distribution of individual criteria ratings (Figure 8) is also valuable to reveal significant shortcomings and challenges. Concerning quantitative evaluation adequacy, expressed by criterion C_3_, the relatively large number of studies in the low-end of the rating spectrum indicates a scarce availability of objective, quantitative measures of image fidelity. This is not unexpected, considering the absence of true reference images when imaging subjects in vivo. Remarkably, the distribution of C_4_ ratings clearly reveals an almost universal absence of evaluation in terms of anatomical verity, quantification capacity, or clinical utility. In other words, image quality was rarely examined in the context of clinical value. This highlights an area with great potential for improvement on the way to more mature evaluation procedures. Another significant observation is that little attention was given to the effect of absorber depth on image quality, as made evident by the ratings for criterion C_5_. However, the highly depth-dependent optical and acoustic attenuation properties of tissue call for more granular assessment, ideally at multiple depth levels, to fully investigate a method’s correction capacity.

### 5.3. Recommendations

In light of the aforementioned, a set of recommendations for the comprehensive evaluation of future research efforts in optoacoustic image quality improvement can be assembled. In the preliminary stage, experiments with numerical and physical phantoms may assess the resolution, existence of artifacts, and overall structural fidelity of the images. This would be greatly facilitated by widely available, standardized targets that cover the entire field of view with absorbers of multiple scales, in a variety of simple and complex geometric arrangements, and situated in realistic, lossy media. Such standardization would provide common references for an objective comparison of different methods, which is currently unfeasible. Absorbing substances with known spectral responses could be involved to enable further validation of the quantification capacity in multispectral configurations. Following the preliminary phase, advancing with evaluation in relevant clinical settings, designed with clinical utility in scope, is preferable. Experiments with one or few individuals can demonstrate possible improvement in clinical utility, especially if findings are reported at multiple depths and scales, and if the enhancement is visually obvious. Nevertheless, studies involving multiple participants and reporting quantitative, statistically significant findings will provide more substantial evidence, less prone to bias.

### 5.4. Limitations

Our work comes with limitations. It can be argued that the current design of the rating procedure does establish an absolutely objective, infallible measure of study quality; an alternative set of weights for the individual criteria or the experiment categories could be proposed, potentially affecting the final selection of studies. Our choices reflect our intent to minimize subjective bias by weighting the primary criteria equally, given that they were elicited from, and were developed to be, applicable to works with a broad range of purposes, as well as to emphasize the importance of evaluation with human subjects. Future efforts could explore alternative designs and the standardization of such sets of criteria to establish thorough study quality control pipelines. Nevertheless, works that have demonstrated evaluation rigor can be reasonably expected to stand out in terms of deviation from their subgroup mean rating, as expressed by the normalized total rating scores. Furthermore, though the exclusive examination of studies that have reported human-involving experiments allowed us to maximize clinical relevance, future work could identify approaches with substantial potential for clinical application by seeking studies that have performed validation with animal subjects in vivo, probing into the domain of pre-clinical optoacoustics research.

### 5.5. Overall Perspective

From an overall perspective, with image quality improvement and assessment approaches in clinical optoacoustic imaging as the core domain of interest, this review retrieved relevant research material through an extensive, systematic search and screening process. An overview of limiting factors that contribute to optoacoustic image quality deterioration was presented; they were categorized into limitations stemming from the hardware, reconstruction algorithms, tissue, and acquisition process. The landscape of the identified image quality improvement approaches was outlined, with concise descriptions of each method’s purpose and key technical details. At a high level, the methods were clustered into signal-domain, image-domain, reconstruction, and hybrid techniques, depending on whether they were introduced prior to, after, or during image reconstruction.

Regarding image quality assessment, proper practices and prevalent shortcomings were sought. In a systematic analysis of the included material, subgroups of studies with common objectives and similarly structured evaluation procedures were composed. The evaluation sections of each study were extensively analyzed, yielding a set of criteria that enabled, inside each subgroup, a comparative rating in terms of evaluation sufficiency. The rating process generated two primary outcomes: first, a selection of works that exhibited significant positive deviation from their subgroup average in terms of total rating was featured, feeding a discussion on characteristics that rendered them eminent, in context with the corresponding criteria. Such characteristics were identified in numerical, physical phantom, and human-involving evaluation experiments. Moreover, concentrating on the latter, the inspection of the criteria-wise distribution of ratings revealed improvement potential in the quantitative assessment; a substantial shortfall in depth-wise evaluation; and a wide disregard of anatomical verity, quantification capacity, and clinical utility aspects.

## 6. Conclusions

Following an initiatory exploration of image-quality-limiting factors and improvement approaches in clinical optoacoustic imaging, our review transitioned to an in-depth, critical analysis in terms of image quality evaluation practices. Given the relative youth of optoacoustic imaging, our findings reflect the naturally experimental, investigative form of a technology still under development. Nevertheless, it is our view that taking them into consideration is going to be integral in the process of advancing the modality towards a more mature state. To this end, we proposed a set of realistic recommendations for the comprehensive and objective evaluation of optoacoustic image quality improvement approaches, with a view to establish solid ground for upcoming research, fostering the transferability of optoacoustic imaging from labs to clinical practice. With respect to future directions, our work may be extended towards preclinical, animal-involving studies. Additionally, keying in on research efforts that have attempted quantitative evaluation in terms of clinical utility could yield a more clinically relevant body of material.

## Figures and Tables

**Figure 2 biosensors-12-00901-f002:**
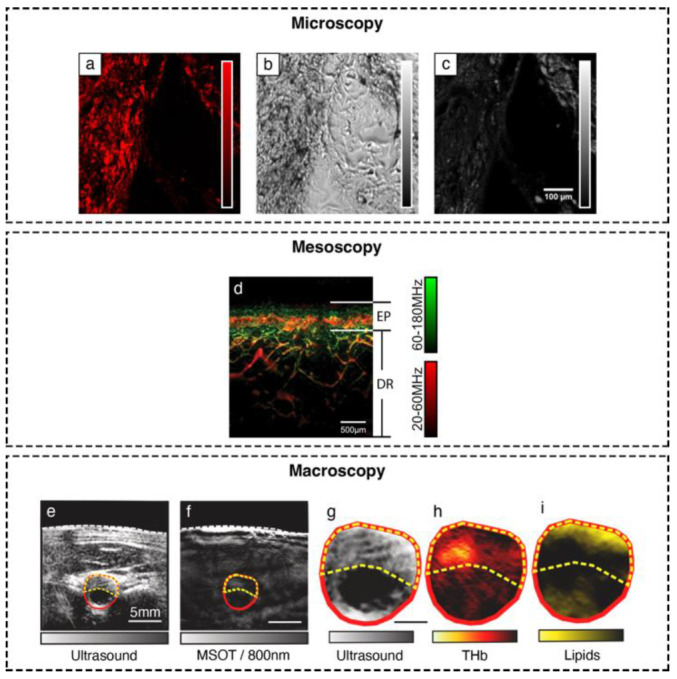
Examples of optoacoustic images. Top: Microscopic images of the lipid core of a human carotid atheroma, adapted from [12] with permission from Elsevier. (**a**) OAM image of embedded red blood cells. (**b**) Brightfield microscopy image. (**c**) Two-photon excitation microscopy fluorescence image. Middle: (**d**) RSOM image of the skin microvasculature on the dorsal aspect of the forearm, adapted from [24], CC BY 4.0 (http://creativecommons.org/licences/by/4.0/, accessed on 26 July 2022). Horizontal lines demarcate the epidermis (EP) and dermis (DR). The color scale represents the size of the imaged microvessels, with red representing large vessels, orange middle-sized vessels, and green small vessels. Bottom: Ultrasound and MSOT images of the carotid artery of a patient with carotid atherosclerosis, adapted from [17] with permission from Elsevier. The arterial lumen is demarcated in red, and the plaque area with a yellow dashed line. (**e**) Ultrasound image. (**f**) MSOT image at 800 nm. (**g**) Magnification of the arterial cross-section in the ultrasound image. (**h**) Magnification of the same cross-section in the spectrally unmixed MSOT image in (**f**), showing the total hemoglobin (THb) signal. (**i**) Same magnification showing the lipids signal.

**Figure 3 biosensors-12-00901-f003:**
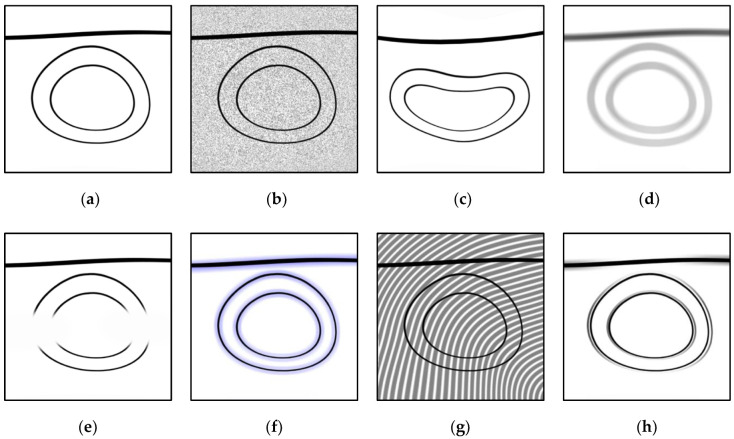
Image domain manifestations of limitations in hardware and algorithms. The upper thick line represents the skin boundary. The circular shapes represent optoacoustic absorbers in tissue. (**a**) Original, unaffected image. (**b**) Background noise due to limited illumination intensity. (**c**) Deformation resulting from acoustic mismatch. (**d**) Loss of high-frequency information (blurring) due to limited bandwidth acquisition. (**e**) Incomplete structure rendering due to limited-view acquisition. (**f**) Negative values (represented with blue) near dominant absorbers. (**g**) Aliasing artifacts. (**h**) Smearing due to inadequate modelling of transducer detector dimensions.

**Figure 4 biosensors-12-00901-f004:**
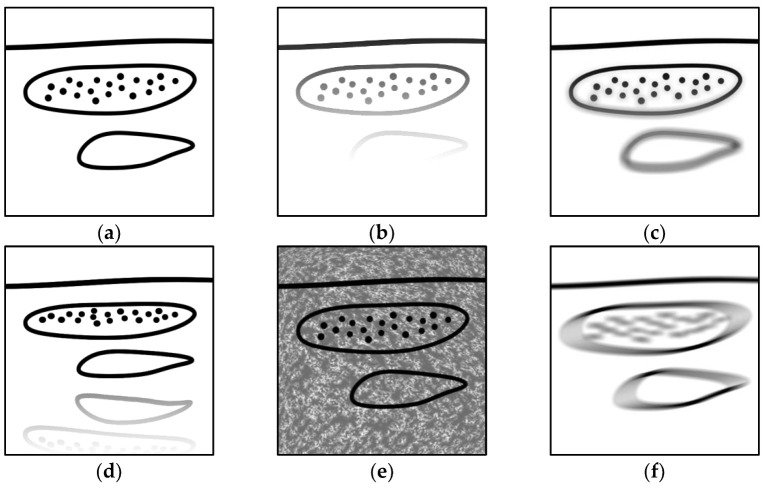
Image domain manifestations of limitations in tissue and in the acquisition process. The upper thick line represents the skin boundary. The shapes below represent optoacoustic absorbers in tissue. (**a**) Original, unaffected image. (**b**) Reduced contrast at depth due to optical attenuation. (**c**) Reduced resolution of deeper structures due to acoustic attenuation. (**d**) Reflection artifacts due to acoustic heterogeneity. (**e**) Clutter originating from optical absorbers outside the imaging plane. (**f**) Directional blurring due to motion.

**Figure 5 biosensors-12-00901-f005:**
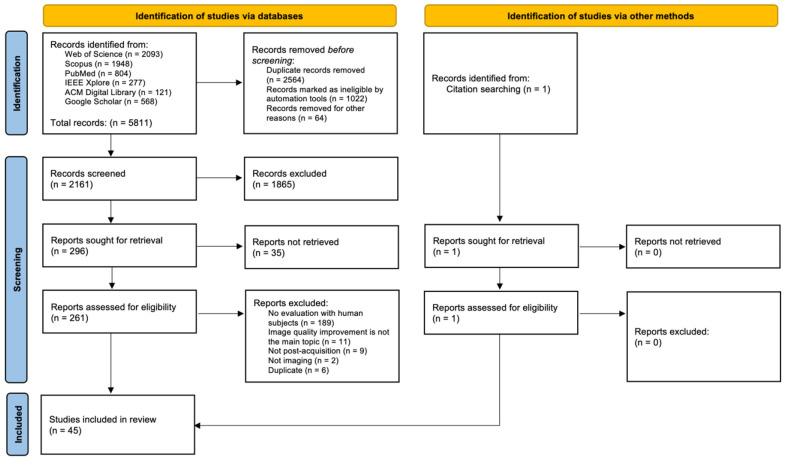
An overview of the literature search and screening procedure. The flow diagram was created according to PRISMA 2020 guidelines for systematic reviews [56].

**Figure 6 biosensors-12-00901-f006:**
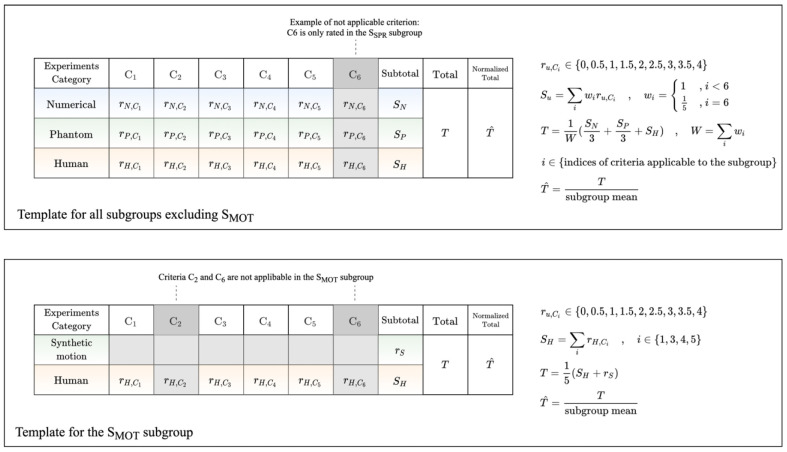
Illustrative overviews of the rating template and procedure for a single study. The relationships between the individual ratings, experiment category-specific subtotals, and total and normalized total ratings are described. **Above**, the overview for all subgroups except for S_MOT_. **Below**, the overview for the S_MOT_ subgroup.

**Figure 7 biosensors-12-00901-f007:**
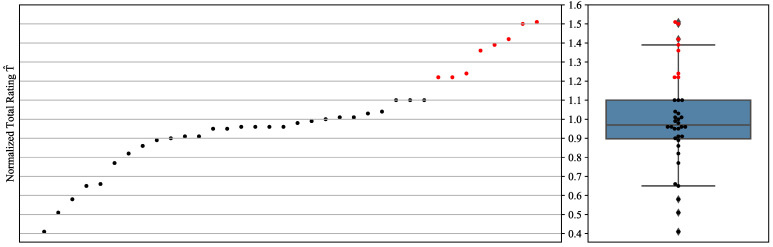
Plots of the normalized total ratings of all studies included in the subgroup analysis. On the **left**, the ratings, sorted in increasing order, are displayed as points. On the **right**, the corresponding box plot with the same points are overlaid. Points that correspond to ratings greater than the upper quartile selection threshold ($Q_{3}=1.1$) are colored red.

**Figure 8 biosensors-12-00901-f008:**
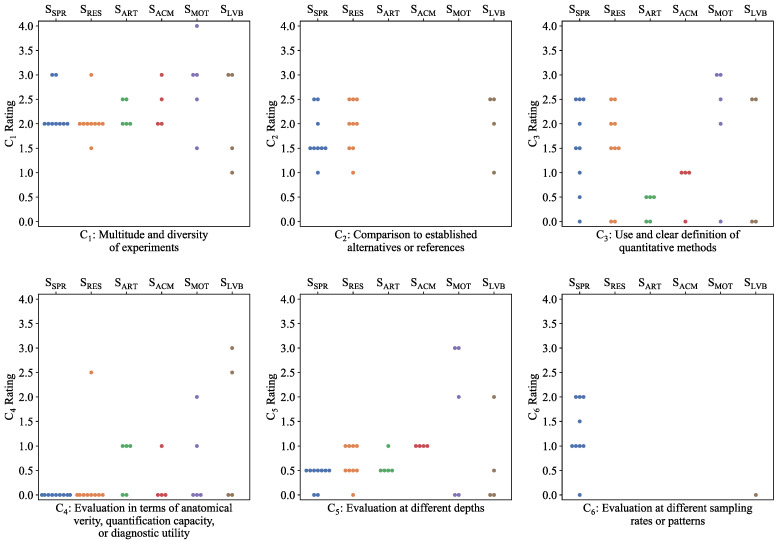
Visual overview of the individual ratings for the human-involving experiments covering all criteria, subgroups, and studies. In case of criteria not applicable to certain subgroups, the corresponding space is left blank. The rating scale is as follows: 0: absent, 1: lacking, 2: adequate, 3: ample, 4: thorough. The points corresponding to different subgroups have been assigned unique colors to facilitate their visual identification.

**Table 1 biosensors-12-00901-t001:** An overview describing the correspondence between primary limiting factors and their manifestations in the image domain. The bullet points (●) indicate correspondence between the limiting factors (table rows), and manifestations (table columns).

	Deformation of Structures	Aliasing Artifacts	Negative Values	Reflection Artifacts (In-Plane)	Clutter	Background Noise
Limited view	●		●			
Limited bandwidth	●	●	●			
Spatial undersampling	●	●				
Inadequate illumination						●
Optical attenuation						●
Out-of-plane absorption					●	
Acoustic attenuation	●		●			
Acoustic heterogeneity	●		●	●	●	
Coupling mismatch	●					
Inexact algorithms	●	●	●			
Motion	●					

**Table 2 biosensors-12-00901-t002:** Summary of the assignment of studies to different subgroups.

Subgroup	Primary Purpose	References
S_SPR_	Improving reconstruction from sparsely sampled measurements	[48,57,58,59,60,61,62,63,64]
S_RES_	Resolution enhancement	[32,44,51,65,66,67,68,69,70]
S_ART_	Elimination of overlaying artifacts	[26,36,71,72,73]
S_MOT_	Motion correction	[40,41,49,74,75]
S_ACM_	Acoustic coupling mismatch modeling	[31,76,77,78]
S_LVB_	Improving reconstruction from limited-view/bandwidth measurements	[50,79,80,81]

**Table 3 biosensors-12-00901-t003:** Summary of the identified criteria, according to which studies were compared and rated.

	Criterion	Description
C_1_	Multitude and diversity of experiments	Captures aspects related to the multitude and diversity of conducted experiments. The reporting of multiple different experiments that investigate the effectiveness of the proposed method from a variety of aspects and through realistic and challenging imaging targets, adds to the validity of the evaluation.
C_2_	Comparison to established alternatives or references	Evaluates the extent to which established methods, known to perform well, are compared to the proposed methods in terms of image fidelity. The applicability of this criterion depends on whether there has been substantial previous activity on the problem the considered subgroup concentrates on. In the case of numerical experiments, the use of ground truth reference images for comparison is considered essential.
C_3_	Use and clear definition of quantitative methods	Apart from the display of reconstructed images, quantitative evaluation using clearly defined methods and metrics is desirable. Studies are expected to quantify the deviation to reference images, when they are available; the display of difference images also helps visualize the spatial distribution of the error magnitude. When possible, metrics can be calculated over sets of multiple images, statistically verifying the reported outcomes.
C_4_	Evaluation in terms of anatomical verity, quantification capacity, or clinical utility	Essentially, the objective of denoising and image quality improvement techniques is to attain increased diagnostic and quantification capacity, advancing the modality towards acceptance in clinical practice. Optoacoustic imaging brings novel opportunities in quantitative imaging, especially through multispectral implementations. In this direction, studies may take the evaluation one step further and provide means to assess the generated images with respect to anatomical or functional plausibility or verity, and, ultimately, clinical utility.
C_5_	Evaluation at different depths	Imaging deep tissue is a major challenge in optoacoustics. Investigating the performance of the compared methods at different tissue depths or distances to the transducer array may reveal otherwise unforeseen limitations.
C_6_	Evaluation at different sampling rates or patterns	This criterion is exclusively applicable in studies that focus on reconstruction from sparsely sampled measurements, where performance may be affected by changes in the sub-sampling rate or the spatial arrangement of active sensors. Evaluations that take this factor into account can be considered as being more comprehensive.

**Table 4 biosensors-12-00901-t004:** Summary of the identified signal domain (pre-processing) approaches. OAT: optoacoustic tomography, MSOT: multispectral optoacoustic tomography, RSOM: raster-scan optoacoustic mesoscopy.

Ref, Title	Modality	Purpose	Subgroup	Year
[27], Ultrasound-guided photoacoustic image reconstruction: image completion and boundary suppression	2D OAT	Correction of incomplete structure reconstruction and suppression of weak absorbers near dominant boundaries due to limited view and bandwidth	-	2013
[82], Adaptive multi-sample-based photoacoustic tomography with imaging quality optimization	2D OAT	Alignment of signals for effective averaging in the presence of motion	-	2015
[83], Spread Spectrum Photoacoustic Tomography With Image Optimization	2D OAT	Recovery of out-of-band information and alignment of signals for effective averaging	-	2016
[72], Identification and removal of laser-induced noise in photoacoustic imaging using singular value decomposition	2D OAT	Elimination of laser-induced noise	S_ART_	2016
[68], Basic study of improvement of axial resolution and suppression of time side lobe by phase-corrected Wiener filtering in photoacoustic tomography	3D OAT	Improvement of axial resolution by time-side-lobes suppression	S_RES_	2018
[40], Motion Quantification and Automated Correction in Clinical RSOM	3D RSOM	Motion correction	S_MOT_	2019
[49], Motion correction in optoacoustic mesoscopy	3D RSOM	Motion correction	S_MOT_	2017
[26], Spatiotemporal Antialiasing in Photoacoustic Computed Tomography	2D OAT	Elimination of spatial aliasing	S_ART_	2020

**Table 5 biosensors-12-00901-t005:** Summary of the identified image domain (post-processing) approaches. OAT: optoacoustic tomography, MSOT: multispectral optoacoustic tomography, OR-PAM: optical-resolution photoacoustic microscopy, RSOM: raster-scan optoacoustic mesoscopy.

Ref, Title	Modality	Purpose	Subgroup	Year
[33], Multiscale multispectral optoacoustic tomography by a stationary wavelet transform prior to unmixing	2D MSOT	Recovery of fine-scale structures masked by spurious negative values	-	2014
[84], 3D Gabor wavelet based vessel filtering of photoacoustic images	3D OR-PAM	Enhancement of vasculature visibility	-	2016
[74], Vascular Registration in Photoacoustic Imaging by Low-Rank Alignment via Foreground, Background and Complement Decomposition	3D OAT	Alignment of shot volumes for effective averaging in presence of motion	S_MOT_	2016
[85], Towards a Fast and Safe LED-Based Photoacoustic Imaging Using Deep Convolutional Neural Network	2D OAT	Improvement of SNR	-	2018
[36], Reflection artifact identification in photoacoustic imaging using multi-wavelength excitation	2D MSOT	Elimination of in-plane reflection artifacts	S_ART_	2018
[73], Feasibility of identifying reflection artifacts in photoacoustic imaging using two-wavelength excitation	2D MSOT	Elimination of in-plane reflection artifacts	S_ART_	2020
[86], Spatial weight matrix in dimensionality reduction reconstruction for micro-electromechanical system-based photoacoustic microscopy	3D OR-PAM	Noise reduction, registration, and deconvolution	-	2020
[29], Negativity artifacts in back-projection based photoacoustic tomography	2D OAT	Elimination of negativity artifacts due to limited view and bandwidth	-	2021
[75], Subpixel and On-Line Motion Correction for Photoacoustic Dermoscopy	3D OR-PAM	Motion correction	S_MOT_	2021
[41], Regional motion correction for in vivo photoacoustic imaging in humans using interleaved ultrasound images	2D OAT	Motion correction	S_MOT_	2021

**Table 6 biosensors-12-00901-t006:** Summary of the identified reconstruction and hybrid approaches. OAT: optoacoustic tomography, MSOT: multispectral optoacoustic tomography.

Ref, Title	Modality	Purpose	Subgroup	Method Class	Method Sub-Class	Year
[70], Coherent-weighted three-dimensional image reconstruction in linear-array-based photoacoustic tomography	3D OAT	Improvement of elevation resolution	S_RES_	Reconstruction	Beamforming	2016
[66], Real-time delay-multiply-and-sum beamforming with coherence factor for in vivo clinical photoacoustic imaging of humans	2D OAT	Improvement of lateral resolution and SNR	S_RES_	Reconstruction	Beamforming	2019
[67], Linear-array photoacoustic imaging using minimum variance-based delay multiply and sum adaptive beamforming algorithm	2D OAT	Improvement of resolution and reduction of sidelobes	S_RES_	Reconstruction	Beamforming	2018
[65], Wave front analysis for enhanced time-domain beamforming of point-like targets in optoacoustic imaging using a linear array	2D OAT	Improvement of lateral resolution and SNR	S_RES_	Hybrid (signal domain analysis, post-processing)	Beamforming	2019
[44], Multiple Delay and Sum With Enveloping Beamforming Algorithm for Photoacoustic Imaging	2D OAT	Suppression of sidelobes and artifacts	S_RES_	Reconstruction	Beamforming	2020
[51], Photoacoustic tomography reconstruction using lag-based delay multiply and sum with a coherence factor improves in vivo ovarian cancer diagnosis	2D OAT	Improvement of resolution and contrast	S_RES_	Reconstruction	Beamforming	2021
[69], Generalized spatial coherence reconstruction for photoacoustic computed tomography	2D OAT	Improvement of resolution, contrast, preservation of relative signal magnitude	S_RES_	Reconstruction	Beamforming	2021
[59], Model-Based Learning for Accelerated, Limited-View 3-D Photoacoustic Tomography	3D OAT	Reconstruction from sub-sampled, limited-view measurements	S_SPR_	Reconstruction	Machine Learning	2018
[58], Approximate k-Space Models and Deep Learning for Fast Photoacoustic Reconstruction	3D OAT	Reconstruction from sub-sampled measurements	S_SPR_	Reconstruction	Machine Learning	2018
[79], Deep-Learning Image Reconstruction for Real-Time Photoacoustic System	2D OAT	Improvement of limited-view and limited-bandwidth reconstruction quality	S_LVB_	Reconstruction	Machine Learning	2020
[80], Y-Net: Hybrid deep learning image reconstruction for photoacoustic tomography in vivo	2D OAT	Improvement of limited-view and limited-bandwidth reconstruction quality	S_LVB_	Hybrid	Machine Learning	2020
[62], High-speed, sparse-sampling three-dimensional photoacoustic computed tomography in vivo based on principal component analysis	3D OAT	Reconstruction from sub-sampled measurements	S_SPR_	Reconstruction	Machine Learning	2016
[61], Dictionary learning sparse-sampling reconstruction method for in-vivo 3D photoacoustic computed tomography	3D OAT	Reconstruction from sub-sampled measurements	S_SPR_	Reconstruction	Machine Learning, Sparsity-based	2019
[60], Compressed sensing photoacoustic tomography in vivo in time and frequency domains	3D OAT	Reconstruction from sub-sampled measurements	S_SPR_	Reconstruction	Sparsity-based	2012
[48], Compressed-sensing photoacoustic computed tomography in vivo with partially known support	3D OAT	Reconstruction from sub-sampled measurements	S_SPR_	Reconstruction	Sparsity-based	2012
[57], Three-dimensional optoacoustic reconstruction using fast sparse representation	3D OAT	Reconstruction from sub-sampled measurements	S_SPR_	Reconstruction	Sparsity-based	2017
[63], Compressed Sensing With a Gaussian Scale Mixture Model for Limited View Photoacoustic Computed Tomography In Vivo	3D OAT	Reconstruction from sub-sampled measurements	S_SPR_	Reconstruction	Sparsity-based	2018
[64], Photoacoustic Reconstruction Using Sparsity in Curvelet Frame: Image Versus Data Domain	3D OAT	Reconstruction from sub-sampled measurements	S_SPR_	Reconstruction	Sparsity-based	2021
[71], Photoacoustic clutter reduction using short-lag spatial coherence weighted imaging	2D OAT	Elimination of clutter	S_ART_	Hybrid (signal domain analysis, post-processing)	Other	2014
[77], Optoacoustic image segmentation based on signal domain analysis	2D MSOT	Accounting for large acoustic mismatches around and inside tissue	S_ACM_	Hybrid (signal domain analysis, reconstruction)	Other	2015
[78], Modeling the variation in speed of sound between couplant and tissue improves the spectral accuracy of multispectral optoacoustic tomography	2D MSOT	Accounting for acoustic mismatch between couplant and tissue	S_ACM_	Reconstruction	Other	2019
[76], Accounting for speed of sound variations in volumetric hand-held optoacoustic imaging	3D OAT	Accounting for acoustic mismatch between couplant and tissue	S_ACM_	Reconstruction	Other	2017
[32], Efficient 3-D Model-Based Reconstruction Scheme for Arbitrary Optoacoustic Acquisition Geometries)	3D OAT	Improvement of resolution, CNR, and overall quality by accounting for finite transducer geometry	S_RES_	Reconstruction	Other	2017
[31], A Synthetic Total Impulse Response Characterization Method for Correction of Hand-Held Optoacoustic Images	2D OAT	Accounting for the total impulse response of the imaging system, including acoustic mismatch between couplant and tissue	S_ACM_	Hybrid (pre-processing, reconstruction)	Other	2020
[81], Soft ultrasound priors in optoacoustic reconstruction: Improving clinical vascular imaging	2D OAT	Improvement of limited-view reconstruction quality	S_LVB_	Reconstruction	Other	2020
[50], Superiorized Photo-Acoustic Non-NEgative Reconstruction (SPANNER) for Clinical Photoacoustic Imaging	2D MSOT	Reconstruction in limited-view and low-SNR settings	S_LVB_	Reconstruction	Other	2021
[87], Opto-Acoustic Image Reconstruction and Motion Tracking Using Convex Optimization	2D MSOT	Simultaneous 3D reconstruction and probe motion tracking	-	Reconstruction	Other	2021

**Table 7 biosensors-12-00901-t007:** The selected studies, their subgroups, and normalized total ratings.

**Reference**	[31]	[44]	[81]	[75]	[59]	[49]	[51]	[50]
**Subgroup**	S_ACM_	S_RES_	S_LVB_	S_MOT_	S_SPR_	S_MOT_	S_RES_	S_LVB_
$\hat{T}$	1.22	1.22	1.24	1.36	1.39	1.42	1.5	1.51

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
