# Peer review of "Image Quality Improvement Techniques and Assessment Adequacy in Clinical Optoacoustic Imaging: A Systematic Review"

_biosensors, 2022, doi:10.3390/bios12100901_

Round 1
Reviewer 1 Report
Authors have reviewed in-vivo optoacoustic imaging studies and carefully provide a criteria-based rating in terms of quality assessment. The review article is very thorough in terms of presenting the issues typically observed in optoacoustic images that degrade the quality, their sources, and image enhancement techniques utilized in in-vivo studies. The systematic analysis to rank the studies in terms of signal or image enhancement, and artifact elimination adds strength to the review process. However, the review on quality assessment techniques and their limitations is not clearly presented. Following are the comments which should be addressed for further improvement of the overall quality of the manuscript.
(1) Line 63, please clarify “expansive range”.
(2) In figure 1, please provide relevant references for the penetration depth and resolution of each modality. Please clarify if the presented values are the obtained range from human study.
(3) In figure 2, the microscopy image need a colorbar, mesocopy require a scale bar.
(4) In Fig. 1, please annotate the circular area at the center within the hypoxic and anoxic ring-shaped area. I would suggest expanding the hypoxic area and classify the overproduced chemicals in terms of the classification mentioned in line 47-49.
(5) In line 129, please clarify “setting robust baseline”. Does it refer to different criteria imposed on the evaluation metrics?
(6) Please clarify, how nonquantitative reconstruction algorithms lead to generate images with negative values. What is nonquantitative reconstruction algorithm?
(7) In Table 2, please clarify the difference between reconstruction from sparse data and limited view data.
(8) Existing quality assessment techniques are not discussed rather authors proposed criteria to rank the reviewed studies which is important and helps researchers to follow specific study protocol. However, this approach does not seem aligned to the title of the paper, specifically, “assessment techniques”.
(9) What are the typical image quality evaluation techniques and their limitations? Are they versatile to be used to compare signal pre-processing or image post-processing techniques? What are the challenges faced in comparing reconstruction algorithms based on the assessment techniques?
(10) Minor errors in sentence structure e.g., line 120-121
Reviewer 2 Report
In this manuscript, the author review the improvement approaches and image quality evaluation practices in clinical optoacoustic imaging in recent years. The author also list the advantages of these methods in tables, which makes the comparison very clear. I think this manuscript can help the researchers in this area to learn the development rapidly and can be publish on Biosensors.
Here are some comments:
1. The Section 3.1 can be a little bit more succinctly.
2. In the tables 4-7, the author uses citation of references to describe the method. I think a brief Introduction or the title of the method can help the readers quickly understand what the author is referring to. For example:
“Alles et al., 2014” canbe “Alles et al., 2014, Photoacoustic clutter reduction using short-lag spatial coherence weighted imaging”
Round 2
Reviewer 1 Report
All the comments and suggestions have been addressed and the article can be accepted in its current form.